# Extraction of Protein from Four Different Seaweeds Using Three Different Physical Pre-Treatment Strategies

**DOI:** 10.3390/molecules25082005

**Published:** 2020-04-24

**Authors:** Jack O’ Connor, Steve Meaney, Gwilym A. Williams, Maria Hayes

**Affiliations:** 1The Food BioSciences Department, Teagasc Food Research Centre, Ashtown, Dublin 15, Ireland; Jack.OConnor@teagasc.ie; 2School of Biological and Health Sciences, Technological University Dublin (TU Dublin)–City Campus, Kevin Street, Dublin 2, Ireland; Steve.Meaney@TUDublin.ie (S.M.); gwilym.williams@TUDublin.ie (G.A.W.)

**Keywords:** seaweeds, proteins, autoclave, high pressure processing, traditional protein extraction, total and free amino acids, solubility

## Abstract

Seaweeds are a rich source of protein and can contain up to 47% on the dry weight basis. It is challenging to extract proteins from the raw biomass of seaweed due to resilient cell-wall complexes. Four species of macroalgae were used in this study-two brown, *Fucus vesiculosus* and *Alaria esculenta*, and two red, *Palmaria palmata* and *Chondrus crispus*. Three treatments were applied individually to the macroalgal species: (I) high-pressure processing (HPP); (II) laboratory autoclave processing and (III) a classical sonication and salting out method. The protein, ash and lipid contents of the resulting extracts were estimated. Yields of protein recovered ranged from 3.2% for *Fucus vesiculosus* pre-treated with high pressure processing to 28.9% protein recovered for *Chondrus crispus* treated with the classical method. The yields of protein recovered using the classical, HPP and autoclave pre-treatments applied to *Fucus vesiculosus* were 35.1, 23.7% and 24.3%, respectively; yields from *Alaria esculenta* were 18.2%, 15.0% and 17.1% respectively; yields from *Palmaria palmata* were 12.5%, 14.9% and 21.5% respectively, and finally, yields from *Chondrus crispus* were 35.2%, 16.1% and 21.9%, respectively. These results demonstrate that while macroalgal proteins may be extracted using either physical or enzymatic methods, the specific extraction procedure should be tailored to individual species.

## 1. Introduction

The worldwide population is set to rise to nine billion by the year 2050 [1]. Looking at consumption trends over the last few decades there has been an increase in daily protein intake and this may be attributed to many different factors one being demand for protein in the sports nutrition sector. Additionally, there has been an increase in demand for vegan/ vegetarian protein alternatives to meat and dairy. Moreover, there has been an increase in the amount of protein consumed especially within developed countries [2]. There is therefore a need to develop robust methods for the extraction of proteins from alternative sources that have a lower ecological impact than those currently used in farming and food processing. People who consume a largely plant-based diet generated through traditional farming methods may consumer larger amounts of herbicide and pesticide residues compared to an omnivorous population [3]. In addition, plant-based diets often struggle to meet the nutritional requirement for essential amino acids and certain vitamins [4]. For example, lysine is typically present at low levels in cereal and root vegetable sourced proteins and blends of cereal and legume proteins are necessary to achieve the correct dietary intake of this amino acid [5,6].

Macroalgae are a rich source of protein and can contain up to 47% on the dry weight basis when harvested and processed from the correct locations and during the correct season [7]. They are classified into three distinct Phyla based on their pigmentation-brown seaweeds (Phylum Phaeophyta), the green seaweeds (Phylum Chlorophyta) and red seaweeds (Phylum Rhodophyta). An advantage of using macroalgae as a protein source is the potential reduction of burden on the environment [2]. Macroalgal cultivation does not use terrestrial lands and does not encroach on traditional farming techniques [3]. The protein yield in tons per hectare per year (ton/ha/yr) for macroalgae is in the range of 2.5–7.5. This value is two to five times higher than that of wheat or legumes. In addition, they can be harvested throughout the year and may be cultivated and some species of macroalgae contain all the essential amino acids required for human nutrition [4]. However, tryptophan and methionine are often limiting amino acids in certain seaweeds. In red algae leucine and isoleucine are the usual limiting amino acid while the brown algae are often deficient in methionine, cysteine and lysine [7,8]. It has been estimated that if just half of the total amount of protein could be extracted more than 20,000 tonnes per year of a new, high-value protein product could be obtained [5].

However, an obstacle to obtaining this protein is the seaweed cell wall. The cell wall complex consists of crystalline cellulose microfibrils that impart structure and which run parallel to the surface of the cell [6]. The cell wall matrix is usually formed from species-specific sulphonated hydrocolloids, e.g., alginates and fucoidan in *Fucus sp*. and in *Alaria esculenta* (which contains up to 37.4% ± 4.0% alginate of the dry matter). Brown algae also contain cellulose, laminarin and mannitol. Red algae contain high amounts of floridean starch, cellulose, d-mannan, d-xylan, and the sulfated galactans; agar and carrageenan. Carrageenan and Furcelleran are found in red algae especially *Chondrus sp* and Xylans are found in *Palmaria sp*. The microfibrils act as a scaffold and are immersed with very little interaction with the hydrocolloid matrix [5,6]. The human gut microbiome or the gastrointestinal (GI) enzymes are unable to break down the cell wall of macroalgae to access the cellular proteins. The protein digestibility of seaweeds is much less compared to digestibility of animal proteins such as casein and whey [7]. Despite this, seaweeds have been used as livestock animal feed in countries including Iceland and the Faroe Islands previously. It was shown previously that gentle heat treatment (70–90 °C) can increase protein digestibility and bio-accessibility in foods due to partial protein denaturation by heat, resolving the tertiary structure and unmasking accession sites for GI enzymes. However, too much heat (>200 °C) can negatively affect protein digestibility and accessibility due to racemization of amino acids from l-to d-amino acids. Only l-amino acids are bioavailable for human consumption [7,8,9,10]. Thus to fully exploit the potential of macroalgae as a protein source, an effective food safe and scalable protein extraction technique is required.

The overall aim of this work was to examine and compare three different physical protein extraction techniques applied to four different seaweeds in order to extract proteins. Much of the research on seaweed proteins to date has looked at the optimum time to harvest seaweeds to obtain the greatest protein yields [8,9]. However, if macroalgae are to be relied on as a protein source, then it may be necessary to have year-round harvest or cultivation of seaweeds which is already common practice for selected seaweeds in Asia and some European countries such as Norway. Extracting macroalgal proteins during seasons when total protein contents are lower than carbohydrates makes breaking the cell wall essential [11]. Three physical methods were evaluated to achieve cell wall breakage: (I) high pressure processing (HPP); (II) an autoclave pre-treatment; and (III) a traditional protein extraction method employing sonication and ammonium sulphate saturation to salt out proteins. The pre-treatment methods are evaluated in terms of their protein extraction efficiency, cost, time and suitability for scale-up when applied to sub-optimal (in terms of protein content) macroalgal material. The three pre-treatments are compared with raw biomass where no pre-treatment was applied and this was used as a control in this work.

## 2. Results

### 2.1. Protein Yields Obtained Following Application of Different Pre-Treatments

The protein content of the generated macroalgal protein extracts isolated from *F. vesiculosus*, *P. palmata*, *C. crispus* and *A. esculenta* using three different pre-treatment strategies were calculated following analysis with the LECO method with tailored nitrogen conversion factors as recommended previously [12,13,14]. Table 1 shows the percentage of protein extracted from the four macroalgal species when exposed to the three pre-treatments. The total quantity of extractable protein ranged from 35.2% ± 3.9% protein for *C. crispus* treated with the classical protein extraction method to 12.5% ± 2.3% protein for *P. palmata* treated using the same method. These results, were in line with expectations as the extraction was carried out in water. The solubility of the native proteins is only approximately 35% in water which sets an upper limit on the amount of protein obtainable via aqueous methods [14,15].

In the case of *F. vesiculosis* the yield of protein obtained ranged from 23.7% ± 2.1% (HPP method) to 35.1% ± 9.1% (classical method). The protein yields obtained for *A. esculenta* ranged from 15.0% ± 2.9% (HPP method) to 18.2% ± 5.3% (classical method) although the HPP method appeared to be more effective for *A. esculenta* than for *F. vesiculosis*. There were no significant differences observed for total extractable protein from either seaweed using the different pre-treatments methods.

The combination of heat and pressure (i.e., autoclave method) was the most effective technique for extraction of protein from *P. palmata*, resulting in a yield of of 21.5% ± 1.4% extractable protein, almost twice that of the classical method. In contrast, to the other red seaweed tests, the classical method applied to *C. crispus* was the most effective showing a significant difference over the other two methods leading to an extractable protein yield of 35.2% ± 3.9%.

### 2.2. Amino Acid Analysis

Seaweed extracts generated using the traditional extraction method, HPP and autoclave pre-treatments were assessed for their total and essential amino acid (EAA) content (Table 2). All extracts generated contained all of the EAA essential for human nutrition. The greatest percentage of EAA–some 43.8%–was found in the protein pellet isolated from *P. palmata* using the classical method. Protein extracted from *C. crispus* using the classical method contained 40.94% EAA while the lowest percent EAA was found in proteins from *F. vesiculosis* generated using the classical protein extraction method (25.45% EAA). Application of the classical extraction method was observed to increase the percentage of the amino acid taurine from 39.72% of total protein in the raw biomass of *F. vesiculosus* to 42.65% taurine of total protein. Glutamic acid was found in all protein extracts and ranged from 12.1% in protein extracts generated using the classical method from *A. esculenta* to 31.12% for the protein extract generated from *P. palmata* using the autoclave method. The percentage and composition of EAAs found in the proteins from *A. esculenta* and *P. palmata* were similar and were richer in threonine, valine, isoleucine, leucine, tyrosine, phenylalanine, histidine, lysine and methionine compared to proteins from *F. vesiculosis* and *C. crispus*.

The traditional protein extraction method, when applied to raw biomass from *P. palmata*, *C. crispus* and *A. esculenta*, increased the percentage EAA by 25–39% EAA. Application of the autoclave pre-treatment to *F. vesiculosus* increased the EAA percent of the protein from 12.5% EAA for raw biomass to 26% for the protein pellet. All protein extracts generated using the traditional protein extraction method resulted in an increase in TAA compared to the raw biomass with the exception of *P. palmata*.

### 2.3. Lipid and Ash Content

The lipid content of the four different seaweeds is shown in Figure 1 and ranged from 0.89% lipid for *A. esculenta* to 4.6% for *F. vesiculosus*. A large proportion of the lipid thus remained in the biomass following application of the different protein extraction methods. The ash content of the protein extracts following HPP ranged from 11.5% for *A. esculenta* to 35.4% for *C. crispus* (Figure 2), while extracts generated using the autoclave method had ash contents ranging from 10.5% for *A. esculenta* to 28.3% *C. crispus*. Classically derived protein extracts contained between 0.34% ash for *A. esculenta* to 4.22% for *C. crispus* (Figure 2).

## 3. Discussion

The autoclave, HPP and classical methods were used to enhance protein extraction from four different seaweeds. The autoclave method was chosen as it combines both heat and pressure. The focus of this paper was on total protein and amino acid composition of extracted proteins for food or bioactive peptide generation. The autoclave pre-treatment method was used previously to extract carbohydrates from seaweeds and was effective at breaking down the cell wall [15]. We used it here and combined it with a centrifugation step to assess if it could enhance protein extraction. The quaternary structure of the protein was not important here. HPP was used, as it is commonly used in the food industry as a preservation method. HPP can burst the cell membranes and walls of both Gram negative and positive microbes, and it was used for this purpose here. The classical method was used, as it is well documented as an effective protein extraction technique when applied to seaweeds. Proximate analysis of the protein extracts identified that, following application of the different extraction methods, the majority of the protein remains in the spent biomass. The classical method was able to recover the greatest percentage of available protein from *C. crispus* (35.2% ± 3.9%). This finding may be explained by the inclusion of a salting out step in the classical method, which would be expected to simultaneously concentrate macroalgal proteins. In addition, in the classical method the biomass is in contact with the extraction solvent for a longer period of time compared to the other methods used. The percentage of EAA present in proteins extracted from all seaweeds using the classical method was greater than the percentage EAA found in proteins extracted using the HPP and autoclave pre-treatments. All pre-treatments, with the exception of the classical protein extraction applied to *P. palmata*, resulted in a greater TAA content in the protein pellet compared to the crude biomass. The differences in the amounts of proteins extracted could be attributed to different cell wall compositions and thereby susceptibility of the seaweeds to cell lysis. It was previously shown that *Sargassum* spp., a brown seaweed, is more recalcitrant to digestion compared to *Gracilaria* spp. and *Ulva* spp., a green seaweed [16].

As protein extractions were carried out in water at neutral pH only certain proteins were extracted into the soluble fraction. To enhance solubility of other proteins and thus overall protein yield, the pH of the extraction solution could be varied, e.g. as described previously. It was shown by Harrysson and colleagues [17] that, due to the solubility of macroalgal proteins at their native pH, typically around 35% of total protein can be extracted by the classical method.

The three pre-treatment methods did not differ significantly in terms of protein yields from the brown seaweeds, while the classical method was most effective at extracting *C.crispus* protein (35.2% ± 3.9%). In contrast, the autoclave method was the most effective at extracting proteins from *P. palmata*. Overall, the autoclave and HPP methods produced similar protein yields from all four seaweeds and protein yields ranged from 17.1–24.3% when the autoclave method was applied to 14.9–23.7% when HPP was used. As noted above, the autoclave method was only significantly more effective in terms of protein yields in the case of *P. palmata,* which is one of the most protein rich macroalgae. The HPP method yielded significantly less protein when applied to the red seaweeds but showed promise for protein extraction from brown seaweeds. Brown seaweed like most of the kelps can be easily grown and harvested in large quantities which would suit the HPP process.

In addition to the efficacy of the methods, the practicalities of each method should also be considered. In terms of time and expense the classical method takes the longest time to perform and uses a large quantity of ammonium sulphate. In contrast, the HPP pre-treatment strategy is fast (three minutes per batch) and cheap (€100 per batch treatment), and suitable for use with large volumes of material. However, while the autoclave method takes longer than HPP it is still faster than the classical method. An additional difference between the methods is related to the equipment-HPP requires specialized and expensive equipment while the classical and autoclave methods may be carried out in most laboratories.

While it is likely that the high temperatures employed during the autoclave pre-treatment could denature proteins, the impact of this on bioavailability of the proteins remains to be evaluated. A study by Evenepoel et al (1998) [18] using a robust isotope-dilution method, revealed that cooked (i.e., denatured) egg proteins were significantly more bioavailable than raw proteins.

The results obtained in this study are similar to protein yields obtained previously from the green seaweeds *Ulvan rigida* and *Ulvan rotanadata* (as described in [19]). The two green seaweeds both were exposed to eight different types of extraction methods and the proteins extracted ranged from <1.00 to 26.8% ± 1.3% for *U. rigida* and 36.1% ± 1.4% for *U. rotandata*. A modified technique used previously which employed an osmotic lysis method was capable of extracting 25.5% of the protein from *Bangiophyceae* and 21.5% from *Phaeophyceae*. This method had the most similarity to the classical method which extracted over 35% protein from *C. crispus* and *F. vesiculosis*. This is likely due to the use of additional steps in the classical method which would have facilitated more degradation of the cell wall such as the sonication and the freeze thaw steps. SDS-PAGE was used previously to determine the sizes of seaweed proteins [20]. In *P. palmata* six proteins with molecular weights of 59.6 kDa, 48.3 kDa, 32.7 kDa, 25.9 kDa, 20.3 kDa and 15.2 kDa were identified. The most abundant proteins had weights of 48.3 kDa, 20.3 kDa and 15.2 kDa. *C. crispus* had seven constant protein bands with molecular weights of 49.3 kDa, 46.2 kDa, 43.2 kDa, 19.8 kDa, 17.2 kDa, 16.4 kDa, and 15.2 kDa [20]. There are high value proteins (130–30,000 $/kg) found in seaweeds including phycobilin proteins. These pigment binding proteins are found in red seaweed. The size of phycoerythrin was identified as 136 kDa previously using SDS PAGE while phycocynin and allophycocynin had bands within the 12–15 kDa range. These proteins are used in industry as food colourings [20].

Ash content was lowest in protein extracts generated using the classical protein extraction method. It is noteworthy that the methods involving pressure resulted in the greatest yield of ash. The ash content of seaweed is high in comparison to terrestrial plant protein extracts which usually contain less than 10% ash with the exception of spinach ash content (20.4%), [21]. The fat content of the protein pellets generated using HPP was the lowest of all extracts generated. These data are consistent with literature values e.g. from Pereira et al. (2011) [22].

Seaweeds can be blended into other plant-based products which lack certain essential amino acids. The main plant based proteins are soybeans and corn. Although soy has high levels of protein, (36 g/100 g) it is deficient in sulphur-containing amino acids like methionine and cysteine. The primary limiting amino acid in corn is lysine [22,23]. In crude *C. crispus,* 4.61% consisted of methionine, and this amino acid was found in all four seaweeds used in this study. A total of 5.58% lysine was found in crude *P. palmata.* The percentage EAA present in the raw seaweed biomass was between 12.77–39.32%. This protein compares favourably to casein which has a reported percentage EAA content of 34% [23]. A common problem for plant-based diets is the lack of the essential amino acid lysine which is found in all four macroalgae extracts at concentrations between 2.10% (HPP treatment of *F. vesiculosus*) and 6.41% (Traditional protein extraction from *P. palmata*). Glutamic acid is found as the prevalent amino acid in protein extracts generated using HPP and the autoclave method. When the traditional protein extraction method is applied to *F. vesiculosus,* the most prevalent amino acid is Taurine (43%). Aspartic acid (10.9–12.1%) and glutamic acid (12.1–12.3%) are the most prevalent amino acids that result in protein extracts following application of the traditional protein extraction method to the three other seaweed species. These results indicate that different protein extraction pre-treatments could be more effective for specific seaweeds. It is evident that there is scope for method optimization to maximize protein yields using the three selected physical pre-treatment methods described here. In addition, non-digestible carbohydrates and phenolics are likely present in the protein extracts and these could inhibit the bioavailability and digestibility of the protein extracts. Future work will involve combining these physical pre-treatments with enzymatic treatments using carbohydrases to further break down the soluble fibers and potential anti-nutrients [24].

## 4. Materials and Methods

### 4.1. Materials

Four species of macroalgae (two brown and two red) were used in this study: *Fucus vesiculosis, Alaria esculenta, Palmaria palmata* and *Chondrus crispus. F. vesiculosus* was harvested in the spring of 2018 and supplied by Shamrock Enterprises Ltd. (Tipperary, Ireland). Both *A. esculenta* and *P. palmata* were collected and harvested by Ocean Sea Vegetables (Sligo, Ireland) in the summer of 2018. *C. crispus* was harvested by Acadian Seaplants Ltd. (Nova Scotia, Canada) in the autumn of 2018. The macroalgae were freeze-dried and subsequently blended and vacuum-packed and stored at −80 °C. All standard chemicals used in this study were obtained from Sigma Aldrich (Sigma Aldrich, Dublin, Ireland); 3.5 KDa SnakeSkin™ was supplied by Thermo Scientific (Thermo Scientific, Ireland) and 100-micron muslin bags were purchased from Armfield Limited (Hampshire, England).

### 4.2. Extraction of Seaweed Proteins Using the Traditional Protein Extraction Method

The traditional protein extraction method was carried out in accordance with the previously published method of Galland–Irmouli [13] with minor modifications. Briefly, 20 g of blended macroalgae was suspended in 1 litre of ultrapure water at 4 °C, to induce cell lysis by osmotic shock. This sample was then sonicated in a sonicator bath (Branson^®^ 3510E-MT Ultrasonic Bath, Branson Ultrasonics, Danbury, CT, USA, 42 KHz) for 1 h at 42 Hz and then frozen at −80 °C for 1 h before thawing in a water bath at room temperature. The remaining insoluble biomass was re-extracted in 0.5 L of ultrapure water as above. The soluble fractions were pooled and protein was salted out by addition of ammonium sulphate to a final saturating concentration of 80% *w*/*v*. The mixture was stirred at 4 °C for 1 h. The sample was then centrifuged at 15,000× *g* using a Sorvall LYNX 6000 super-speed centrifuge (Thermo Fisher Scientific, Dublin, Ireland) for 1 h at 4 °C. The protein pellet was then re-suspended in a minimal volume of ultrapure water and put into 3.5 kDa dialysis tubing and suspended in 5 L of ultrapure water with stirring for 24 h. The pellet was recovered, frozen and freeze-dried. All samples were dried using a freeze-drying protocol in an FD80GP freeze-drier (Cuddon Freeze Dry, New Zealand). The freeze-drying program used was as follows: −20 °C for 5 h; 10 °C for 5 h; 37.5 °C for 1 h; 12.5 °C to 25 °C for 95 h with the end temperature being 25 °C.

### 4.3. Extraction of Seaweed Proteins Using High Pressure Processing (HPP)

Twenty grams of each macroalgae were suspended in 1 L of ultrapure water and divided and decanted into four 250 mL polyethylene terephthalate HPP compatible bottles (The Packaging Centre Limited, Fox and Geese House, Naas Road, Dublin 22, Ireland).

Samples were placed in a Hiperbaric 420 High-Pressure Processor (HPP Tolling, St. Margaret’s Dublin, Ireland) and exposed to 600 MPa of pressure for four minutes. Separation of soluble and insoluble material was achieved by passing the post-HPP treated material through a 100-micron muslin bag (Armfield Limited, Bridge House, West Street, Ringwood, BH24 1DY, England). Water-soluble protein extracts and spent biomass were frozen, freeze-dried as above and stored at −80 °C until required.

### 4.4. Extraction of Seaweed Proteins Using an Autoclave Pre-Treatment

Twenty grams (×3) of macroalgae were suspended individually in 1 L of ultrapure water. Samples were placed in an autoclave (Astell ASB260 BT) at a temperature of 124 °C, 0.101 Mpa for 2 × 15 min cycles. Samples were removed and allowed to subsequently cool to room temperature. The soluble portion fraction was separated from the insoluble biomass using a 100 micron muslin bag as described earlier. The soluble and insoluble fractions were frozen and freeze-dried as described earlier and subsequently stored at −80 °C until further use.

### 4.5. Proximate Analysis

The protein content of extracted samples and spent biomass was determined using a LECO FP628 protein analyser based on the Dumas method which uses a furnace to measure total nitrogen and according to AOAC method 992.15, 1990. The nitrogen to protein conversion factors used for individual seaweeds were: *F. vesiculosus*: 4.30; *A. esculenta*: 4.45, P. *palmata*: 4.10 and *C. crispus*: 3.55. Lipid was quantified using the smart system 5 microwave moisture drying oven and NMR Smart Trac rapid Fat Analyser using AOAC Official Methods 985.14 and 985.26, 1990. Ash was determined using a Carbolite Muffle Furnace overnight at 600 °C according to method ISO 2171.

### 4.6. Total Amino Acid Determination

Determination of the total and free amino acid composition of the protein extracts generated was determined. Protein extracts were hydrolysed using 6 M HCL at 110 °C for 23 h, as described previously by Hill et al (1965), [25]. The samples were then de-proteinized by mixing equal volumes of 24% (*w*/*v*) tri-chloroacetic acid and sample. These were allowed to stand for 10 min at room temperature before centrifugation at 14,400× *g* for 10 min. The supernatants were removed and diluted with 0.2 M sodium citrate buffer, pH 2.2 to give approximately 250 nmol of each amino acid residue. Samples were then diluted 1 in 2 with the internal standard norleucine to give a final concentration of 125 nm/mL. Amino acids were quantified using a Jeol JLC-500/V amino acid analyser (Jeol (UK) Ltd., Herts, Garden city, UK) fitted with a Jeol Na^+^ high performance cation exchange column.

### 4.7. Statistical Analysis

In this study, all measurements were carried out in triplicate. All statistical analysis was performed using SAS version 9.4 analysed with a One-way ANOVA using the GLM procedure and having the means compared with the Tukey test.

## 5. Conclusions

The results from this work demonstrate that the application of physical pre-treatment methods can yield protein extracts with enhanced EAA contents compared to the raw biomass. The protein yields were similar to those shown in other studies focused on extraction except these pre-treatments can be up-scaled. The results also show that different pre-treatment strategies should be applied to individual seaweeds in order to maximize the yield of protein as in the case of *P. palmata* where the greatest yield of protein from this seaweed was obtained using the autoclave method. Future work will involve the use of enzymatic hydrolysis tailored to the individual seaweeds in conjunction with one of the pretreatment methods used.

## Figures and Tables

**Figure 1 molecules-25-02005-f001:**
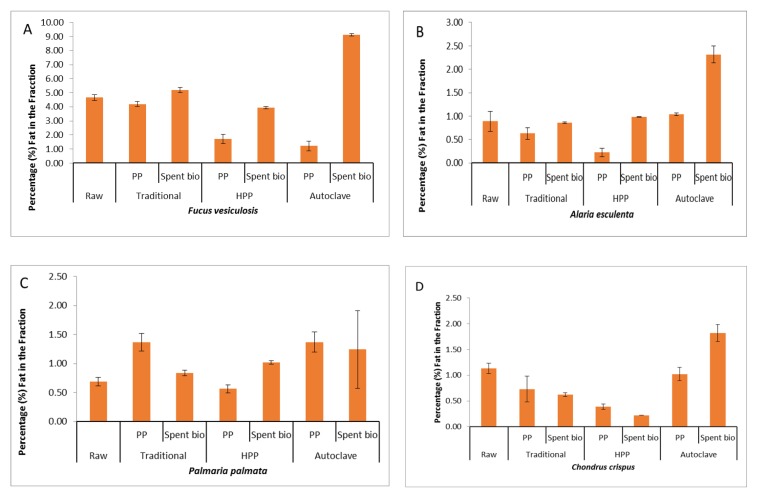
**A**–**D**: Percentage (%) lipid in raw, protein pellets and spent biomass of macroalgae.

**Figure 2 molecules-25-02005-f002:**
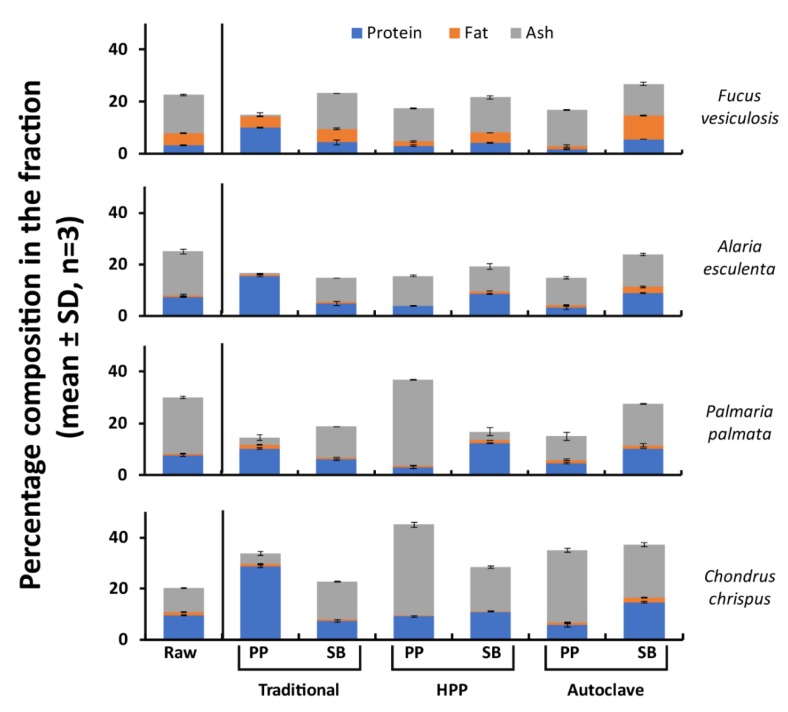
Proximate analysis of each seaweed species by pre-treatment method. Raw = raw biomass, PP = protein pellet, SB = spent biomass.

**Table 1 molecules-25-02005-t001:** The percentage of extractable protein obtained from different macroalgae treated using the classical protein extraction method, high pressure processing (HPP) or autoclave pre-treatment methods.

Species	Percentage of Extracted Protein Obtained Following Pre-Treatment Application (mean ± SD, n = 3)			
	*F. vesiculosis*	*A. esculenta*	*P. palmata*	*C. crispus*
Classical	35.1 ± 9.1	18.2 ± 5.3	12.5 ± 2.3	35.2 ± 3.9 *
HPP	23.7 ± 2.1	15.0 ± 2.9	14.9 ± 1.1	16.1 ± 0.5
Autoclave	24.3 ± 1.5	17.1 ± 1.5	21.5 ± 1.4 *	21.9 ± 3.3

Methods which resulted in a significant difference in percentage yield of protein for each macroalga specie are indicated by * (*p* < 0.05); *n* = 3.

**Table 2 molecules-25-02005-t002:** Amino acid profile of the crude macroalgae and macroalgal derived protein extracts (percentage of the Total amino acids), N=1.

Amino Acids	*Fucus vesiculosis*				*Alaria esculenta*				*Palmaria palmata*				*Chondrus crispus*			
	Crude	Trad	HPP	Auto	Crude	Trad	HPP	Auto	Crude	Trad	HPP	Auto	Crude	Trad	HPP	Auto
Threonine	1.51	2.59	4.09	4.57	4.63	5.79	4.54	3.56	4.71	4.89	3.67	3.66	4.57	5.54	2.82	3.86
Valine	1.23	2.53	4.33	3.69	5.41	6.63	3.34	3.71	6.15	7.39	4.17	3.94	3.69	6.25	3.74	4.37
Isoleucine	1.20	1.64	2.22	1.89	4.31	4.54	1.59	1.43	3.64	4.59	3.11	2.15	1.89	4.51	2.34	3.12
Leucine	0.60	3.25	4.13	2.71	7.09	7.27	2.32	1.56	5.91	7.73	3.83	3.68	2.71	6.93	3.23	4.98
Tyrosine	ND	4.63	0.77	0.97	0.13	2.14	0.00	0.24	0.49	2.11	0.00	0.15	0.97	2.68	4.54	0.17
Phenylalanine	0.77	2.27	3.27	2.47	4.58	4.67	2.95	2.50	3.84	4.91	2.61	2.66	2.47	4.29	0.40	4.49
Histidine	5.70	4.45	2.90	1.47	5.04	2.28	5.26	3.33	4.65	2.57	5.26	2.95	1.47	2.15	3.06	3.55
Lysine	1.75	2.44	2.10	4.16	5.28	5.22	0.96	1.65	5.58	6.40	3.22	3.67	4.16	5.31	4.14	4.58
Methionine	NA	1.66	1.84	4.61	2.85	3.46	0.99	2.30	2.70	3.17	0.80	2.34	4.61	3.29	0.83	2.59
Cysteic acid	5.11	2.27	4.53	4.04	3.24	1.93	2.54	2.64	4.13	1.75	8.09	6.31	4.04	2.94	3.45	2.74
Taurine	39.72	42.65	6.62	11.86	4.34	1.93	8.71	6.08	3.04	1.43	7.22	2.66	11.86	1.20	18.30	10.26
Aspartic acid	4.70	5.67	12.97	12.99	9.86	12.11	10.86	10.16	10.27	10.98	9.79	11.86	12.99	12.09	7.15	9.17
Serine	0.72	2.21	3.18	3.72	5.14	4.59	3.28	3.80	5.07	4.31	3.32	3.91	3.72	5.16	2.93	4.58
Glutamic acid	17.09	5.57	28.77	25.63	14.09	12.11	27.12	33.89	15.15	12.28	24.70	31.12	25.63	12.13	25.83	21.74
Glycine	1.69	2.19	5.16	4.21	5.28	6.63	4.94	3.77	5.80	6.14	6.20	5.74	4.21	5.22	7.22	5.42
Alanine	4.63	3.30	6.82	5.18	6.32	7.65	14.70	14.55	6.33	7.88	4.73	6.07	5.18	7.55	4.20	5.32
Cysteine	8.93	8.57	3.65	3.30	0.64	0.96	3.83	1.70	2.17	1.05	2.78	0.76	3.30	0.71	1.26	0.39
Arginine	1.20	2.12	2.66	2.52	7.00	5.88	2.05	1.63	5.96	6.32	2.87	2.98	2.52	6.48	2.77	5.09
Proline	3.44	12.23	3.06	1.72	4.78	4.22	NA	1.50	4.40	4.10	3.64	3.39	1.72	5.58	1.79	3.60
∑EAA(%)	12.77	25.45	25.65	26.55	39.32	41.99	21.96	20.27	37.68	43.76	26.66	25.20	26.55	40.94	25.10	31.70
TAA (mg/g)	44.30	57.82	17.60	19.72	61.12	93.70	31.34	37.62	112.18	73.10	35.23	59.68	137.41	226.26	73.40	

TAA: Total amino acids, ∑EAA: The sum of the Essential Amino Acids; Crude = dried seaweed (whole); HPP = High pressure processing; Trad = traditional or classical extraction method; Auto = autoclave method.

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
