# Peer review of "Extraction of Protein from Four Different Seaweeds Using Three Different Physical Pre-Treatment Strategies"

_molecules, 2020, doi:10.3390/molecules25082005_

Round 1
Reviewer 1 Report
Many researchers are addressing extraction methods which, indeed, influence yield of different chemical compounds. The authors address specifically protein extraction from seaweeds which have a hard cell wall, difficult to break, and thus challenging to increase metabolites yield.
Pre-treatment is an interesting approach, because this enhances cell walls disruption and it may be performed by mechanical, chemical, thermal, or enzymatic pre-treatment. Also mixed methods have been widely used.
So the methods suggested are not new, although as far as I could see, autoclaving has been assessed mainly for lipid extraction and high pressure homogenization (as well as ultrasonication) for protein solubilization, but mainly for microalgae.
Dumas method was used to assess total protein, which is a suitable short assay method for measuring proteins in seaweeds. The conversion factors used are also adequate for the studied species.
Lipid and ash methods are also standard quantification methods.
Yet, heat treatment application to protein is rather undesirable, for protein are highly sensitive to temperature, causing their denaturation. So, using this method for protein assessment must be explained.
Also, the choice of the pre-treatment other methods should be explained. There are so many possibilities, why choose this three in particular?
Results present one figure that cannot be understood, for there is no explanation in the text what “PP” and “Spent-bio” mean.
Besides, fat is in a totally different percentage, thus it should be in a different figure.
The discussion is poor. There are plenty of information regarding results from other techniques and pre-treatments used for the same species. There should be comparisons from other authors to understand if these are good results.
Besides, there should be comparisons with other commonly used food species such as soybean, because in the introduction there are statements regarding vegetarianism.
Standard deviations are not shown but for the first table, so we do not know if replicas were made, and this is a major flaw.
Author Response
Response to reviewer comments_April 2020 -Paper Molecules 764008
Reviewer 1 comments
Many researchers are addressing extraction methods which, indeed, influence yield of different chemical compounds. The authors address specifically protein extraction from seaweeds which have a hard cell wall, difficult to break, and thus challenging to increase metabolites yield. Pre-treatment is an interesting approach, because this enhances cell walls disruption and it may be performed by mechanical, chemical, thermal, or enzymatic pre-treatment. Also mixed methods have been widely used. So the methods suggested are not new, although as far as I could see, autoclaving has been assessed mainly for lipid extraction and high pressure homogenization (as well as ultrasonication) for protein solubilization, but mainly for microalgae. Dumas method was used to assess total protein, which is a suitable short assay method for measuring proteins in seaweeds. The conversion factors used are also adequate for the studied species. Lipid and ash methods are also standard quantification methods. Yet, heat treatment application to protein is rather undesirable, for protein are highly sensitive to temperature, causing their denaturation. So, using this method for protein assessment must be explained. Also, the choice of the pre-treatment other methods should be explained. There are so many possibilities, why choose this three in particular?
Response to reviewer 1 comments: We agree with the reviewer. The original paper lacked an explanation concerning the selection of the pre-treatment methods. We have amended the revised paper to reflect the reasons why HPP, autoclave and the classical method were selected for use in this paper. This is described in the discussion section lines 172-182 as follows: “The autoclave, HPP and classical methods were used to enhance protein extraction from four different seaweeds. The autoclave method was chosen as it combines both heat and pressure. The focus of this paper was on total protein and amino acid composition of extracted proteins for food or bioactive peptide generation. The autoclave pre-treatment method was used previously to extract carbohydrates from seaweeds and was effective at breaking down the cell wall. We used it here and combined it with a centrifugation step to assess if it could enhance protein extraction. The quaternary structure of the protein was not important here. HPP was used as it is commonly used in the food industry as a preservation method. HPP can burst the cell membranes and walls of both gram negative and positive microbes and it was used for this purpose here. The classical method was used as it is well documented as an effective protein extraction technique when applied to seaweeds.”
Reviewer 1, comment 2: Results present one figure that cannot be understood, for there is no explanation in the text what “PP” and “Spent-bio” mean. Besides, fat is in a totally different percentage, thus it should be in a different figure.
Response to reviewer 1 comments: We agree with the reviewer and we have now included an explanation on the figure for the abbreviations PP, Spent-bio (Lines 158 and 159).
Reviewer 1, comment 3: The discussion is poor. There are plenty of information regarding results from other techniques and pre-treatments used for the same species. There should be comparisons from other authors to understand if these are good results.
Besides, there should be comparisons with other commonly used food species such as soybean, because in the introduction there are statements regarding vegetarianism.
Response to reviewer 1, comment 3: We again agree with the reviewer and we have altered the discussion text significantly to reflect these comments. We have included references related to soy and egg proteins and the discussion now reads as follows, lines 218-221“
While it is likely that the high temperatures employed during the autocalve pre-treatment could denature proteins, the impact of this on bioavailability of the proteins remains to be evaluated. A study by Evenepoel et al (1998) (29) using a robust isotope-dilution method, revealed that cooked (i.e. denatured) egg proteins were significantly more bioavailable than raw proteins”.
We have included the following reference also: “Evenepoel, P., Geypens, B., Luypaerts, A., Hiele, M., Ghoos, Y., Rutgeerts, P. (1998). Digestibility of cooked and raw egg protein in humans as assessed by stable isotope techniques. J. Nutr. 128, 10, 1716-1722.)”
Also we have included soy in the text, lines 237-241: “Seaweeds are a source of complete protein with abundant essential amino acids. They can be blended into other plant-based products which lack certain essential amino acids. The main plant based proteins are soybeans and corn. Although soy has high levels of protein it is deficient in Sulphur containing amino acids like Methionine and Cysteine. The primary limiting amino acid in corn is lysine (27).”
Reviewer 1, comment 4: “Standard deviations are not shown but for the first table, so we do not know if replicas were made, and this is a major flaw”
Response: We agree again with the reviewer and we have added details concerning statistical packages used and include the replicate numbers on the tables (n=3) and we have included standard deviations in the text now for yields.
Reviewer 2 Report
The current study focuses on the extraction of protein from four different seaweeds using three different physical pre-treatment methods. The paper seems original the experiments are well designed, and the conclusions were confirmed by the results. I recommend considering the paper after revising the following points:
- A space is required between number and unit except for %. Please revise in the whole manuscript
- It’s interesting to make a comparison with casein (amino acid composition). However, I recommend referring to suggested patterns of AA requirement (mg per g of protein) from FAO/WHO. For more details see table 4 in the paper:
https://doi.org/10.1007/s10722-011-9739-9
- In figure 1 please correct Latin name to italic
- The statistical analysis is required in table 2 and figure 1. So, we can confirm the conclusions; the significant differences between extractions method and the most efficient one.
- The references style seems inappropriate please revise according to the journal instructions
Author Response
Reviewer 2
The current study focuses on the extraction of protein from four different seaweeds using three different physical pre-treatment methods. The paper seems original the experiments are well designed, and the conclusions were confirmed by the results. I recommend considering the paper after revising the following points:
Reviewer 2 comment 1: A space is required between number and unit except for %. Please revise in the whole manuscript.
Response: We have checked the document and inserted a space between numbers and units except for %.
Reviewer 2, comment 2: It’s interesting to make a comparison with casein (amino acid composition). However, I recommend referring to suggested patterns of AA requirement (mg per g of protein) from FAO/WHO. For more details see table 4 in the paper:
https://doi.org/10.1007/s10722-011-9739-9
Response: We have referenced the link below table 2 in the revised paper to reflect this.
Reviewer 2, comment 3: In figure 1 please correct Latin name to italic
Response: We have corrected the Latin name to italics in table 2.
Reviewer 2 comment 4: “The statistical analysis is required in table 2 and figure 1. So, we can confirm the conclusions; the significant differences between extractions method and the most efficient one”
Response: We have presented the statistical analysis in Figure 1 and table 2.
The references style seems inappropriate please revise according to the journal instructions
Reviewer 3 Report
In this study, the authors describe three different pre-treatment strategies to extract proteins from four seaweeds. Their main purpose of this study was to examine and compare three different physical protein extraction techniques in the four different seaweeds and to find different pretreatment strategies that are effectively applicable to individual seaweeds to maximize the yield of protein. Seaweeds are a very potent source of protein and for biomedical applications. By this time, many techniques including protein extraction have been established for seaweeds.
However, in my point of view, the presentation of the techniques of this study is very basic. I think many reports have already existed in the literature of this similar study. Are the three techniques newly established in this study? If not, what are the differences in this study with other similar studies?
The discussion is very poorly written. I would like to see a brief discussion which could support to establish their findings. The authors should conclude their results applying three techniques are unique.
Also, to support the results, the authors might purify the proteins and apply molecular studies (SDS-PAGE) to see the molecular weight of proteins for each sample. Alternatively, authors could discuss with other similar studies from the literature which have already studied in these aspects.
Page 1, lines 15-16: “Four species of macroalgae were used in this study - two brown, Fucus vesiculosus and Alaria esculenta, and two red, Palmaria palmata and Chondrus crispus (red)”- (red) should be deleted.
Page 1, lines 19-21: “Yields of protein recovered ranged from 3.2% for Fucus vesiculosus pre-treated with HPP to 28.9% protein recovered for Chondrus crispus treated with the classical method”. The extraction of protein yields of all samples should be presented here to justify and conclude the results.
Author Response
Reviewer 3
Reviewer 3 comment: In this study, the authors describe three different pre-treatment strategies to extract proteins from four seaweeds. Their main purpose of this study was to examine and compare three different physical protein extraction techniques in the four different seaweeds and to find different pretreatment strategies that are effectively applicable to individual seaweeds to maximize the yield of protein. Seaweeds are a very potent source of protein and for biomedical applications. By this time, many techniques including protein extraction have been established for seaweeds. However, in my point of view, the presentation of the techniques of this study is very basic. I think many reports have already existed in the literature of this similar study. Are the three techniques newly established in this study? If not, what are the differences in this study with other similar studies?
Response: We agree that the extraction of proteins using pre-treatments is not a new study. This study however is the first to show the use of scalable methods applied to seaweeds from 3 different Phyla for protein extraction. We have shown that different pre-treatments work better for selected seaweeds – and that these methods are scalable.
Reviewer 3 comment: The discussion is very poorly written. I would like to see a brief discussion which could support to establish their findings. The authors should conclude their results applying three techniques are unique.
Response: We have revised the discussion substantially from the original draft and have included comparisons with soy and egg. We have highlighted how our methods are unique and why they were chosen and we have concrete conclusions concerning the yields of protein obtained using different pretreatments.
Reviewer 3 comment: Also, to support the results, the authors might purify the proteins and apply molecular studies (SDS-PAGE) to see the molecular weight of proteins for each sample. Alternatively, authors could discuss with other similar studies from the literature which have already studied in these aspects.
Response: The authors were interested only in protein yields measured by Dumas. A next stage will involve SDS-PAGE analysis. We also carried out initial SDS-PAGE and got varying protein sizes from this and have included this in the text.
Reviewer 3, comment: Page 1, lines 15-16: “Four species of macroalgae were used in this study - two brown, Fucus vesiculosus and Alaria esculenta, and two red, Palmaria palmata and Chondrus crispus (red)”- (red) should be deleted.
Response: We have deleted (red) as instructed by reviewer 3.
Reviewer 3 comment: Page 1, lines 19-21: “Yields of protein recovered ranged from 3.2% for Fucus vesiculosus pre-treated with HPP to 28.9% protein recovered for Chondrus crispus treated with the classical method”. The extraction of protein yields of all samples should be presented here to justify and conclude the results.
Response: We have included the extraction of proteins yields of all samples on page 1.
Round 2
Reviewer 1 Report
The reviewed paper has been widely corrected and is now much clearer. Thus, no major corrections are required. Although some recommendations I made before were not corrected, thus I’m sending them again for further review.
Table 1 has some mistake in the last column; I believe data are missing in the first part (before it showed data for casein).
Also, figure 1 has been corrected but it cannot be read by itself, because fat must be in a different scale in order to be properly distinguishable. I think two figures are required.
Line 57. The comment previously made has not been corrected: Again, this is not true for many seaweeds. Although present, methionine and tryptophan are often limiting amino acids.
Line 64: The comment previously made has not been corrected: Include information on the Phycocolloids for all the species studied.
Line 69: The comment previously made has not been corrected: you refer it in the keywords, but not in the introduction. Because cell wall is not digested, protein digestibility much lower in seaweeds than in other organisms such as animals. You should discuss this further comparing bioavailability in different organisms, so you can compare later the total protein content.
Line 74: The comment previously made has not been corrected: Besides harvest wild seaweeds an interesting choice is the cultivation that already occurs in many European and Asian countries, and this doesn’t depend on season so much. You can refer this as well.
Line 99 to line 102. And in both cases statistical significat differences were found.
Line 111. The comment previously made has not been corrected: “This compares favorably to casein which has a reported percentage EAA content of 45.1% (19).” should go to discussion.
Line 113. Table 2 refers that C. crispus from traditional methods shows 40.94%, not 40.24%. Which one is correct?
Line 121 and 122: are rich in these amino acids compared to what?
Line 153: “These data are consistent with literature values e.g. from Pereira et al. (2011)” should go to discussion.
Line 196 – I agree with the statement, but do you have any scientific support for the different composition of cell wall of the studied species?
Line 328 – Hill et al. date is missing. Also, You must explain, somewhere, that the three pre-treatment methods are compared, in the results, with raw biomass, with no pre-treatment, and that this is your control.
Author Response
Response to reviewer 1 comments_Molecules 764008
Paper Molecules_764008_resubmission & corrections
Reviewer 1 comments: Table 1 has some mistake in the last column; I believe data are missing in the first part (before it showed data for casein).
Also, figure 1 has been corrected but it cannot be read by itself, because fat must be in a different scale in order to be properly distinguishable. I think two figures are required.
Response: We apologies to reviewer 1 for this oversight. We have included table 1 to include casein. In addition, we have included a figure for fat – we now have figures 1.1 and 1.2 in the paper.
The fat figure is inserted as follows:
Figure 1.1 A-D: Percentage (%) in raw, protein pellets and spent biomass of macroalgae
Reviewer 1 comments: Line 57. The comment previously made has not been corrected: Again, this is not true for many seaweeds. Although present, methionine and tryptophan are often limiting amino acids.
Response: We have corrected the text to reflect the fact that methionine and tryptophan are often limiting amino acids in seaweeds. Section now reads as:
“An advantage of using macroalgae as a protein source is the potential reduction of burden on the environment (2). Macroalgal cultivation does not use terrestrial lands and does not encroach on traditional farming techniques (3). According to Krimpen and colleagues (2013) the protein yield in tons per hectare per year for macroalgae is in the range of 2.5-7.5 ton/ha/yr. This value is 2-5 times higher than that of wheat or legumes. In addition, they can be harvested throughout the year and some species of macroalgae contain all the essential amino acids required for human nutrition (4). However, many algae have one or two limiting amino acids; for example tryptophan and methionine are often limiting amino acids in seaweeds. In red algae leucine and isoleucine are the usual limiting amino acid while the brown algae are often deficient in methionine, cysteine and lysine (30). It has been estimated that if just half of the total amount of protein could be extracted more than 20,000 tonnes per year of a new, high-value protein product could be obtained (5).”
Reviewer 1 comments_Line 64: The comment previously made has not been corrected: Include information on the Phycocolloids for all the species studied.
Response: We agree with the reviewer and we have edited the text. We now believe that sufficient information is provided concerning phycocolloids. The focus of the paper is protein. The section now reads;
“However, an obstacle to obtaining this protein is the seaweed macroalgal cell wall. The cell wall complex consists of crystalline cellulose microfibrils that impart structure and which run parallel to the surface of the cell (6). The cell wall matrix is usually formed from the more species-specific sulphonated hydrocolloids, e.g. alginates and fucoidan in Fucus sp. and in Alaria esculenta (contains up to 37.4% (±4.0) alginate of the dry matter). Brown algae also contain cellulose, laminarin and mannitol. Red algae contain high amounts of floridean starch, cellulose, D-mannan, D-xylan, and the sulfated galactans; agar and carrageenan. Carrageenan and Furcelleran are found in red algae especially Chondrus sp and Xylans are found in Palmaria sp. The microfibrils act as a scaffold and are immersed with very little interaction with the hydrocolloid matrix (5,6).”
Reviewer 1 comment _Line 69: The comment previously made has not been corrected: you refer it in the keywords, but not in the introduction. Because cell wall is not digested, protein digestibility much lower in seaweeds than in other organisms such as animals. You should discuss this further comparing bioavailability in different organisms, so you can compare later the total protein content.
Response: We have added more information in this section to address the concerns of reviewer 1 concerning bioavailability. The section now contains information on seaweed V animal protein bioavailability and reads:
“The human gut microbiome or the gastrointestinal (GI) enzymes are unable to break down the cell wall of macroalgae to access the cellular proteins. The protein digestibility of seaweeds is much less compared to digestibility of animal proteins such as casein and whey (7).Despite this, seaweeds have been used as livestock animal feed in countries including Iceland and the Faroe Islands previously (32,8). It was shown previously that gentle heat treatment (70 -90°C) can increase protein digestibility and bioaccessibility in foods due to partial protein denaturation by heat, resolving the tertiary structure and unmasking accession sites for GI enzymes. Too much heat (>200°C) can negatively affect protein digestibility and accessibility due to racemization of amino acids from L- to D-amino acids. Only L-amino acids are bioavailable for human consumption (7). Thus to fully exploit the potential of macroalgae as a protein source, an effective food safe and scalable protein extraction technique is required.”
Reviewer 1 comment_Line 74: The comment previously made has not been corrected: Besides harvest wild seaweeds an interesting choice is the cultivation that already occurs in many European and Asian countries, and this doesn’t depend on season so much. You can refer this as well.
Response: We agree with reviewer 1 and have addressed their concerns and have included cultivation in the text. The text now reads: “However, if macroalgae are to be relied on as a protein source then it may be necessary to have year-round harvest or cultivation of seaweeds which is common practice in Asia and in some European countries such as Norway. Extracting macroalgal proteins during seasons when total protein contents are lower than carbohydrates makes breaking the cell wall essential (10).”
Reviewer 1 comment_Line 99 to line 102: And in both cases statistical significant differences were found.
Response: Yes, in both cases statistically significant differences were found and we have edited the text accordingly. The text now reads: The combination of heat and pressure (i.e. autoclave method) was the most effective technique for extraction of protein from P. palmata, resulting in a yield of of 21.5 ±1.4 % extractable protein, almost twice that of the classical method. In contrast, to the other red seaweed tests, the classical method applied to C. crispus was the most effective showing a significant difference over the other two methods leading to an extractable protein yield of 35.2 ±3.9%.
Reviewer 1 comments_Line 111: The comment previously made has not been corrected: “This compares favorably to casein which has a reported percentage EAA content of 45.1% (19).” should go to discussion.
Response: We agree with the reviewer and we have moved this text to the discussion section and is now found on line 270-271 in the discussion section.
Reviewer 1_comment_Line 113. Table 2 refers that C. crispus from traditional methods shows 40.94%, not 40.24%. Which one is correct?
Response: We agree with reviewer 1 and we have corrected this. The value is 40.94% and is shown on line 125 of the paper.
Reviewer 1 comments_Line 121 and 122: are rich in these amino acids compared to what?
Response: We have corrected the text and it now reads as: “The percentage and composition of EAAs found in the proteins from A. esculenta and P. palmata were similar and were richer in threonine, valine, isoleucine, leucine, tyrosine, phenylalanine, histidine, lysine and methionine compared to those found in the F. vesiculosis and C. crispus.”
Reviewer 1 comments_Line 153: “These data are consistent with literature values e.g. from Pereira et al. (2011)” should go to discussion.
Response: We agree with the reviewer and we have moved this line to the discussion section lines 259.
Reviewer 1 comments_Line 196: – I agree with the statement, but do you have any scientific support for the different composition of cell wall of the studied species?
Response: We have included a reference for this statement.
Reviewer 1 comment_Line 328 – Hill et al. date is missing. Also, You must explain, somewhere, that the three pre-treatment methods are compared, in the results, with raw biomass, with no pre-treatment, and that this is your control. We have included this in the introduction lines 93-94: “The three pre-treatments are compared with raw biomass where no pre-treatment was applied and this was used as a control in this work.”
Response: We have included the reference for Hill et al (Hill et al., 1965) and explained in the materials and methods that the three pre-treatment are compared with raw biomass where no pre-treatment was applied and this was our control.

Reviewer 3 Report
The paper is improved. However, there are still some errors remain in the revised version and they should be fixed for getting the acceptance.
Page 1, lines 14-18: “Seaweeds are a rich source of protein and can contain up to 47% on a dry weight. It is challenging to extract proteins from the raw biomass of seaweed due to its resilient cell-wall complex. Seaweeds are a rich source of protein and can contain up to 47% on a dry weight basis. It is challenging to extract proteins from the raw biomass of seaweed due to its resilient cell-wall complex.” It seems the authors mistakenly wrote the sentences two times. Also, ---on a dry weight. It should be written ---on the dry weight. I would suggest to the authors to check the English errors throughout the manuscript.
The authors mentioned that they carried out initial SDS-PAGE and got varying protein sizes from their samples and have included the results in the text. I couldn’t find these results in the revised version. As I mentioned in my previous review comments if the authors didn’t purify the proteins using SDS-PAGE, they could discuss with other similar studies from the literature which have already studied in these aspects. If they do so, the readers will get a clear idea about their extracted proteins.
Author Response
Molecules_revised submission 764008
Reviewer 2 comment: Page 1, lines 14-18: “Seaweeds are a rich source of protein and can contain up to 47% on a dry weight. It is challenging to extract proteins from the raw biomass of seaweed due to its resilient cell-wall complex. Seaweeds are a rich source of protein and can contain up to 47% on a dry weight basis. It is challenging to extract proteins from the raw biomass of seaweed due to its resilient cell-wall complex.” It seems the authors mistakenly wrote the sentences two times. Also, on a dry weight. It should be written ---on the dry weight. I would suggest to the authors to check the English errors throughout the manuscript.
Response: We have revised the paper in accordance with the guidance of reviewer 1. We have deleted the duplication text and we have written on the dry weight basis as suggested by the reviewer. We hope that this is now satisfactory. The section now reads:
“Seaweeds are a rich source of protein and can contain up to 47% on the dry weight. It is challenging to extract proteins from the raw biomass of seaweed due to its resilient cell-wall complex. Four species of macroalgae were used in this study - two brown, Fucus vesiculosus and Alaria esculenta, and two red, Palmaria palmata and Chondrus crispus. Three treatments were applied individually to the macroalgal species: (I) high-pressure processing; (II) laboratory autoclave processing and (III) a classical salting out method. The protein, ash and lipid content of the resulting extracts were estimated. Yields of protein recovered ranged from 3.2% for Fucus vesiculosus pre-treated with HPP to 28.9% protein recovered for Chondrus crispus treated with the classical method. The yields of protein recovered using the classical, HPP and autoclave pre-treatments applied to Fucus vesiculosus were 35.15, 23.7% and 24.3%, respectively; yields from Alaria esculenta were 18.2%, 15.0% and 17.1% respectively; yields from Palmaria palmata were 12,5 %, 14,9% and 21.5% respectively and finally yields from Chondrus crispus were 35,2%, 16,1% and 21.9%, respectively. These results demonstrate that while macroalgal proteins may be extracted using either physical or enzymatic methods, the specific extraction procedure should be tailored to individual species. “
Reviewer 2 comment: The authors mentioned that they carried out initial SDS-PAGE and got varying protein sizes from their samples and have included the results in the text. I couldn’t find these results in the revised version. As I mentioned in my previous review comments if the authors didn’t purify the proteins using SDS-PAGE, they could discuss with other similar studies from the literature which have already studied in these aspects. If they do so, the readers will get a clear idea about their extracted proteins.
Response: We have included some text concerning SDS-PAGE and the protein sizes by comparison with the literature. This is found in the text on lines: “SDS-PAGE was used previously to determine the sizes of seaweed proteins. In P. palmata six proteins with molecular weights of 59.6 kDa, 48.3 kDa, 32.7 kDa, 25.9 kDa, 20.3 kDa and 15.2 kDa were identified. The most abundant proteins had weights of 48.3 kDa, 20.3 kDa and 15.2 kDa. C. crispus had seven constant protein bands with molecular weights of 49.3 kDa, 46.2 kDa, 43.2 kDa, 19.8 kDa, 17.2 kDa, 16.4 kDa,and 15.2 kDa (19). There are high value proteins (130-30000 $/kg) found in seaweeds including phycobilin proteins. These pigment binding proteins are found in red seaweed. The size of phycoerythrin was identified as 136 kDa previously using SDS PAGE while phycocynin and allophycocynin had bands within the 12-15 kDa range. These proteins are used in industry as food colourings (20).”
